# High-Order Sinc-Correlated Model Vortex Beams

**Jixian Wang** [1,2]**, Zhangrong Mei** [2,]*[ID]**, Yonghua Mao** [3]**, Xiaohui Shi** [1,2] **and Guoquan Zhou** [1][ID]

1 Department of Optical Engineering, Zhejiang A&F University, Hangzhou 311300, China
2 School of Electronic Information, Huzhou College, Huzhou 313000, China
3 Department of Physics, Huzhou University, Huzhou 313000, China
* Correspondence: meizr@zjhu.edu.cn

**Abstract:** We propose a new partially coherent vortex source model in which the spatial correlation function is a sinc function on the difference from the $q$-th power of the coordinates of two points of the source field. The beam radiated by such source is termed the high-order sinc-correlated model vortex (SCMV) beam. We derived the propagating formula of the cross-spectral density (CSD) function for SCMV beams in atmospheric disturbances. On the basis of the derived analytical expression, the behavior of the spectral density of the SCMV beams propagating in free space and atmosphere turbulence was investigated under comparative analysis. The results show that the spectral densities of such beams exhibited interesting novel features, which were significantly different from those of the trivial vortex beams.

**Keywords:** partially coherent beams; vortex beams; atmospheric turbulence; propagation characteristics

## 1. Introduction

Light fields with different spectral coherence characteristics generated by random sources will produce very different propagation characteristics [1]. Therefore, to generate radiation fields with specific properties, modeling the spatial correlation function of random sources has become a research hotspot in recent decades. As a classical partially coherent beams model, the Gaussian Schell model source is well known for its Gaussian coherence and spectral density [2]. The Schell model type is currently the foundation for the majority of investigations on the structure of the partially coherent light source. Since Gori et al. established the necessary requirements for a valid CSD function [3,4], much consideration and discussion have been given to the studies of the spatial correlation structure of partially coherent sources, and a number of partially coherent beam models with extraordinary correlation structures have been put forth by later researchers [5,6]. Furthermore, it was discovered that special correlated structured beams are superior to conventional correlated structured beams in terms of their many distinctive transmission qualities, such as non-uniform correlated Schell model (NUCSM) beams with a self-focused, self-shifted effect in transmission [7,8]; multi-Gaussian Schell model (MGCSM) beams with circularly symmetric flat-topped intensity distribution in the far field [9–11]; and cosine Gaussian Schell model (CGSM) sources form a dark hollow light intensity distribution after transmission over a distance [12]. Two types of scalar random beams introduced in [13] can be generated in the far field with annular intensity.

When the light beams are applied to some real scene such as optical tracking, imaging, laser radar [14–16], and so on, they will inevitably interact with the medium. However, the quality of the beam will be degraded due to the interaction of the light field with the medium during propagation, and the development of the beam in real applications will be hindered by this. Therefore, the research on the light field's propagation characteristics in random media and how to improve the anti-interference ability of light beams in propagation has become an important topic. In recent years, this field has been widely studied [17–22].

Recently, a newly developed modeling approach was put forth in [23]. Instead of being the difference between the two source points, as in the Schell model, the coherence function in the two directions $x$ and $y$ is depicted as a separable function. On the basis of this approach, we propose a new class of vortex beams, called SCMV beams, in this paper. We analyzed the distribution of their spectral densities when they propagate in free space and turbulent atmosphere, and we also investigated the effect of turbulence on the spectral densities and profiles at different turbulence parameters, obtaining some interesting results.

## 2. Sinc-Correlated Model Vortex Beams

Reviewing coherence theory, for a typical planar source, we usually elaborate on the optical field's coherence using the CSD function, which reveals the degree of correlation between two points in the optical field.

$$W_0(\rho_{10}, \rho_{20}) = \langle U^*(\rho_{10}) U(\rho_{20}) \rangle, \tag{1}$$

where $\rho_{10}$ and $\rho_{20}$ are the position vectors of two points on the source plane, the start is the complex conjugate, and the ensemble average of the wave field $U$ marked with $\langle \cdots \rangle$. According to the superposition principle, we know that a true CSD function must satisfy the nonnegative limiting condition. Thus, the CSD must be expressed as follows in the integral form [3]:

$$W_0(\rho_{10}, \rho_{20}) = \int p(\mathbf{v}) H_0^*(\rho_{10}, \mathbf{v}) H_0(\rho_{20}, \mathbf{v}) \mathrm{d}^2 \mathbf{v}. \tag{2}$$

in which $H_0$ is a complex-valued function that explains how a light field correlates. The contour of the correlation function of a light field is defined by $p(\mathbf{v})$, a nonnegative function that can be Fourier converted. As an example of a typical class, let us choose a kernel $H_0$, which is a Fourier-like form dependent on $\mathbf{v}$ [3],

$$H(\rho, \mathbf{v}) = \tau(\rho) \exp[-i2\pi \mathbf{v} \cdot g(\rho)], \tag{3}$$

where $\tau(\rho)$ is a possible beam intensity amplitude function in parallel with $g(\rho)$, which is a real vector function, so the CSD function can be presented in this general form:

$$W_0(\rho_{10}, \rho_{20}) = \tau^*(\rho_{10}) \tau(\rho_{20}) \widetilde{p}[g(\rho_{10}) - g(\rho_{20})], \tag{4}$$

where $\widetilde{p}$ represents the Fourier converted form of $p(\mathbf{v})$. Assume that $p(\mathbf{v})$ is a formal function that can be separated. Create it, for instance, as the product of the $x$ and $y$ directional functions:

$$p(\mathbf{v}) = \delta_x^q \mathrm{rect}\left(v_x \delta_x^q\right) \delta_y^q \mathrm{rect}\left(v_y \delta_y^q\right). \tag{5}$$

where $\mathrm{rect}(x)$ is the rectangular function, $\delta_x^q$ and $\delta_y^q$ are power functions of positive constant $\delta$ that has the length dimension, and we set the Laguerre–Gaussian model for the complex amplitude function $\tau(\rho)$,

$$\tau(\rho) = \left(\frac{\rho}{\sigma}\right)^l \exp\left(-\frac{\rho^2}{\sigma^2}\right) \exp(il\phi). \tag{6}$$

where $\exp(il\phi)$ is the phase factor, and $l$ indicates the topological charge. In this paper, the value of the topological charge is taken as $l = 1$.

On substituting Equations (5) and (6) into Equation (4), the CSD function expression can be written as follows:

$$\begin{aligned} W_0(\rho_{10}, \rho_{20}) = {} & \left(\frac{\rho_{10}\rho_{20}}{\sigma^2}\right)^l \exp\left(-\frac{\rho_{10}^2 + \rho_{20}^2}{\sigma^2}\right) \exp[il(\phi_{20} - \phi_{10})] \\ & \times \mathrm{sinc}\left(\frac{x_{10}^q - x_{20}^q}{\delta_x^q}\right) \mathrm{sinc}\left(\frac{y_{10}^q - y_{20}^q}{\delta_y^q}\right), \end{aligned} \tag{7}$$

where $\mathrm{sinc}(x)$ is normalized, and $\mathrm{sinc}(x) = \sin(\pi x)/(\pi x)$ except for $\mathrm{sinc}(0) = 1$. The CSD represented by Equation (7) satisfies the nonnegative definiteness condition. Thus, the source is physically realizable. This model serves as an example of a novel type of partially coherent vortex light sources that we refer to as SCMV beams.

### 3. Cross-Spectral Density of the SCMV Beams Propagating in Atmospheric Turbulence

Next, we focus on the beam's propagation characteristics produced by this light source. Assuming that the beam propagates in the turbulent medium-filled paraxial region of a $z > 0$ half-space, according to the diffraction theory of the light field, we can obtain the CSD function characterizing the relationship between any two positions $(\rho_1, z)$ and $(\rho_2, z)$ in the optical radiation field [24]:

$$W(\rho_1, \rho_2, z) = \left(\frac{k}{2\pi z}\right)^2 \iint W_0(\rho_{10}, \rho_{20}) \exp\left\{-\frac{ik}{2z}\left[(\rho_1 - \rho_{10})^2 - (\rho_2 - \rho_{20})^2\right]\right\}$$
$$\times \langle \exp[\Psi^*(\rho_1, \rho_{10}, z) + \Psi(\rho_2, \rho_{20}, z)]\rangle_M \mathrm{d}^2\rho_{10}\mathrm{d}^2\rho_{20}. \tag{8}$$

In this equation, the optical field's wave number $k = 2\pi/\lambda$, which depends on the wavelength $\lambda$. The phase correlation term $\langle \exp[\Psi^*(\rho_1, \rho_{10}, z) + \Psi(\rho_2, \rho_{20}, z)]\rangle_M$ represents the perturbation caused by the random medium. It can be well approximated as

$$\langle \exp[\Psi^*(\rho_1, \rho_{10}, z) + \Psi(\rho_2, \rho_{20}, z)]\rangle_M$$
$$= \exp\left\{-\frac{\pi^2 k^2 z}{3}\left[(\rho_1 - \rho_2)^2 + (\rho_1 - \rho_2)(\rho_{10} - \rho_{20}) + (\rho_{10} - \rho_{20})^2\right]\int_0^\infty \kappa^3 \varphi(\kappa)d\kappa\right\}. \tag{9}$$

Then, we can obtain the CSD

$$W(\rho_1, \rho_2, z) = \left(\frac{k}{2\pi z}\right)^2 \iint W_0(\rho_{10}, \rho_{20}) \exp\left\{-\frac{ik}{2z}\left[(\rho_1 - \rho_{10})^2 - (\rho_2 - \rho_{20})^2\right]\right\}$$
$$\times \exp\left\{-\frac{\pi^2 k^2 z}{3}\left[(\rho_1 - \rho_2)^2 + (\rho_1 - \rho_2)(\rho_{10} - \rho_{20}) + (\rho_{10} - \rho_{20})^2\right]\right. \tag{10}$$
$$\left. \times \int_0^\infty \kappa^3 \varphi(\kappa)d\kappa\right\}\mathrm{d}^2\rho_{10}\mathrm{d}^2\rho_{20}.$$

where $\varphi(\kappa)$ is a function that determines the intensity of the turbulent medium and is named as the power spectrum [25]. In our work, we choose the function as

$$\varphi(\kappa) = A(\alpha)\widetilde{C}_n^2 \frac{\exp(\kappa^2/\kappa_m^2)}{(\kappa^2 + \kappa_0^2)^{\alpha/2}}, \ 0 \le \kappa < \infty, \ 3 < \alpha < 4, \tag{11}$$

which represents the power spectrum in the non-Kolmogorov turbulence model. The term $\widetilde{C}_n^2$ is the refractive index structural parameter in unit $m^{3-\alpha}$. In addition, $\kappa_0 = 2\pi/L_0$ and $\kappa_m = c(\alpha)/l_0$. The external and internal scales of turbulence are denoted by $L_0$ and $l_0$, respectively.

$$c(\alpha) = \left[\Gamma\left(5 - \frac{\alpha}{2}\right)A(\alpha)\frac{2\pi}{3}\right]^{1/(\alpha-5)}, \tag{12}$$

$$A(\alpha) = \Gamma(\alpha - 1)\frac{\cos(\alpha\pi/2)}{4\pi^2}, \tag{13}$$

with $\Gamma(x)$ being the gamma function. Substituting Equation (7) into Equation (10), the CSD is derived as

$$W(\rho_1, \rho_2, z) = \left(\frac{k}{2\pi z\sigma}\right)^2 \exp\left[-\frac{ik}{2z}(\rho_1^2 - \rho_2^2) - \frac{\pi^2 k^2 z}{3}(\rho_1 - \rho_2)^2\right]$$
$$\times \left\{\iint x_{10}x_{20}F_x\mathrm{d}x_{10}\mathrm{d}x_{20}\iint F_y\mathrm{d}y_{10}\mathrm{d}y_{20} - i\iint x_{10}F_x\mathrm{d}x_{10}\mathrm{d}x_{20}\iint y_{20}F_y\mathrm{d}y_{10}\mathrm{d}y_{20}\right. \tag{14}$$
$$\left. + i\iint x_{20}F_x\mathrm{d}x_{10}\mathrm{d}x_{20}\iint y_{10}F_y\mathrm{d}y_{10}\mathrm{d}y_{20} + \iint F_x\mathrm{d}x_{10}\mathrm{d}x_{20}\iint y_{10}y_{20}F_y\mathrm{d}y_{10}\mathrm{d}y_{20}\right\},$$

where

$$
\begin{aligned}
F_j &= \exp\left[-\frac{\rho_{10}^2+\rho_{20}^2}{\sigma^2} - \frac{ik}{2z}\left(\rho_{10}^2 - \rho_{20}^2 - 2\rho_1\rho_{10} + 2\rho_2\rho_{20}\right)\right] \\
&\quad \times \exp\left\{-\frac{\pi^2 k^2 z}{3}\left[(\rho_1-\rho_2)(\rho_{10}-\rho_{20}) + (\rho_{10}-\rho_{20})^2\right]\int_0^\infty \kappa^3 \varphi(\kappa)\mathrm{d}\kappa\right\} \\
&\quad \times \mathrm{sinc}\left(\frac{j_{10}^q - j_{20}^q}{\delta_j^q}\right), \quad j = x, y.
\end{aligned}
\tag{15}
$$

On the basis of the propagation expression of the CSD function expressed in Equation (14), we can calculate the spectral density of the SCMV beam in free space and in non-Kolmogorov turbulence using the spectral density definition:

$$
S(\rho, z) = W(\rho, \rho, z).
\tag{16}
$$

## 4. Numerical Results

We now numerically integrate Equation (12) to study the propagation characteristics of the SCMV beams in free space and atmospheric turbulence. Unless as otherwise indicated in the captions, the source parameters values are set to $\sigma = 1$ cm, $\delta_x = \delta_y = 7$ mm, $\lambda = 632.8$ nm. The external and internal scales of turbulence are set to $L_0 = 1$ m, $l_0 = 1$ mm. In Figure 1, we show the evolution of the longitudinal spectral density distribution of the beam propagating from the source plane to 250 m corresponding to different values of $q$. Due to the presence of the vortex structure, it made the spectral density of the radiation field produced by the beam show a pattern of dark hollow distribution. Meanwhile, some noteworthy phenomenon emerged here. When $q$ was odd, the size of this spectral density dark hollow distribution profile gradually became smaller as the beam transmission distance increased, and when the beam continued to be transmitted to a certain distance, this dark hollow distribution pattern completely disappeared and the central intensity of the beam was no longer zero. However, the difference was that when $q$ was even, the spectral density dark hollow contour was always able to exist stably throughout the propagation of the beam. That is, when $q$ was odd, the SCMV beam had a vortex structure that disappeared after a certain distance during the transmission, while the vortex structure was able to exist stably during propagation when $q$ was even. In addition, we also found that the dark hollow region of the beam exhibited a self-focusing effect during propagation when $q$ was even. Moreover, the beam profile narrowed at the focus position, and the location where the focusing effect occurred varies for different values of $q$. This result was similar to the results exhibited by the self-focusing vortex beam [26]. From this result, it can be seen that by varying the $q$ value, a possibility for beam shaping is provided.

Figure 2 depicts the transverse spectral density distribution of the SCMV beam propagating in free space at $z = 100$ m. As can be seen from the figure, when $q$ was even, it had a central intensity distribution of zero as the general vortex beam, but unlike the intensity of the ordinary vortex beam that presented a rotationally symmetric distribution, the spectral density distribution of the SCMV beam showed a pattern of four bright spots surrounding the central dark core. In the case where $q$ took odd values, the spectral density distribution of the beam no longer had a central dark core, as shown in Figure 2a–c. The $q = 1$ showed a circularly symmetric distribution, while the intensity of the beam converged toward the center along the $x$ and $y$ axes for larger values of $q$, and the central intensity was distributed more rapidly from a hollow to a solid distribution.

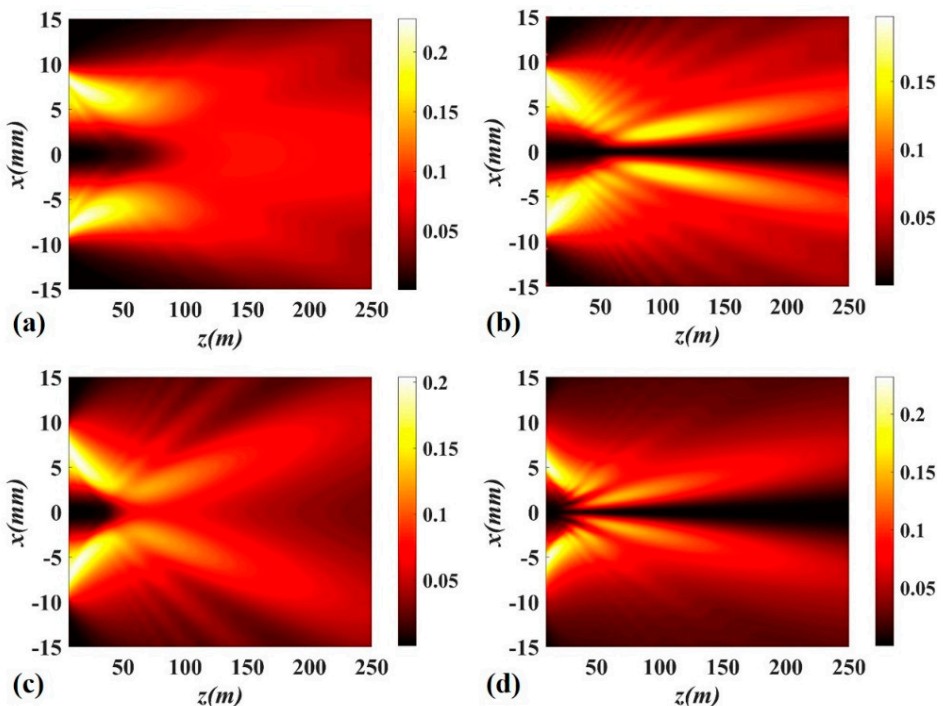

**Figure 1.** The longitudinal spectral density distribution generated by a source defined in Equation (7) with a variety of *q* values in free space. (**a**) *q* = 1; (**b**) *q* = 2; (**c**) *q* = 3; (**d**) *q* = 4.

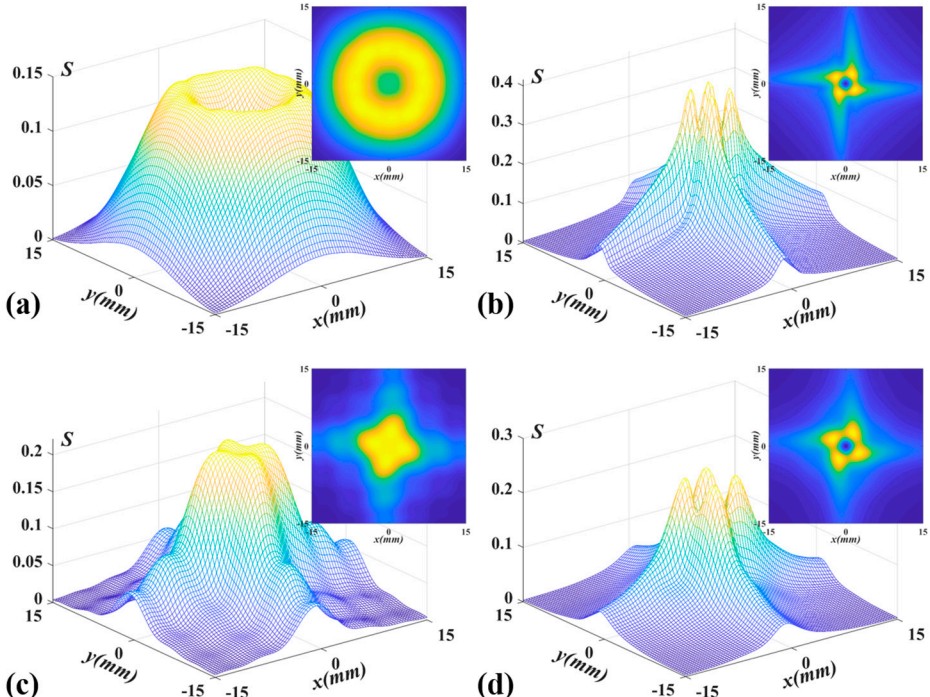

**Figure 2.** The distribution of transverse spectral density of SCMV at distance *z* = 100 m in free space corresponding to Figure 1. (**a**) *q* = 1; (**b**) *q* = 2; (**c**) *q* = 3; (**d**) *q* = 4.

Next, we turned our attention to exploring the variation of the SCMV beam during propagation. As a representative example, we worked on the case of *q* = 2. Figure 3 depicts the distribution of the intensity at some specified distance planes as the beam propagated in free space. Comparing Figure 3a,b, we can clearly see that the size of the dark hollow profile of the beam shrunk significantly with the increase in the transmission

distance, and the focusing effect appeared at 100 m, while the spectral density distribution gradually converged toward the $x$ and $y$ axes. The four bright spots tightly surrounded the central optical axis and had a tendency to rotate, and the maximum value of light intensity during propagation also appeared here. As the distance continued to increase, the beam profile gradually expanded, the focusing effect disappeared, and the beam profile gradually evolved to a similar appearance to that of the near-field region.

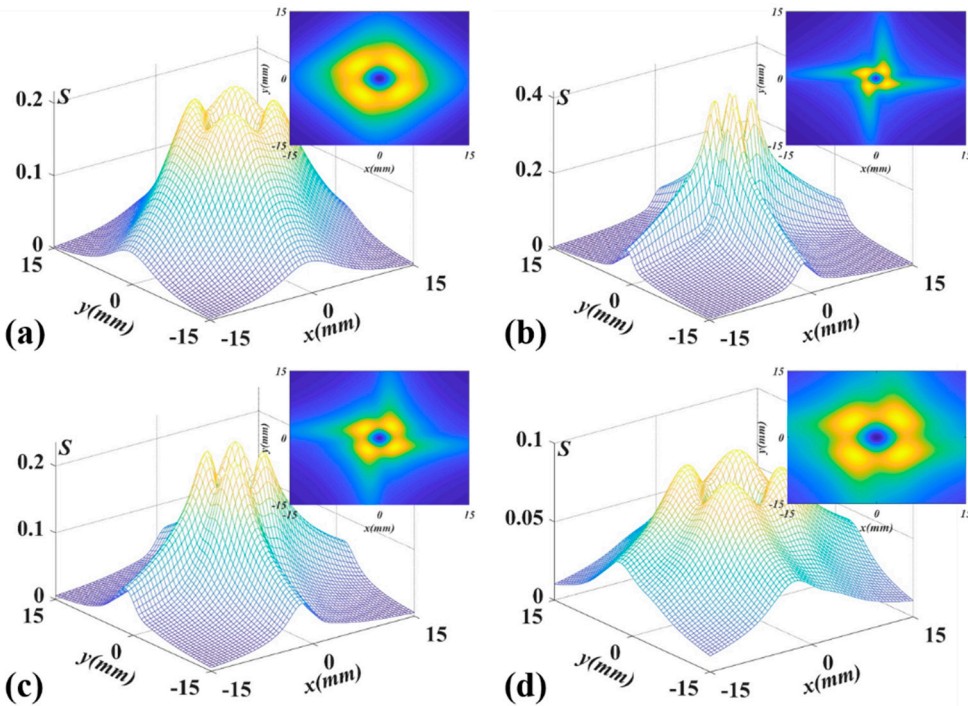

**Figure 3.** The distribution of intensity as the beams travelled through free space for the case of $q = 2$ at some specified distance. (**a**) $z = 50$ m; (**b**) $z = 100$ m; (**c**) $z = 150$ m; (**d**) $z = 250$ m.

Then, we investigated the effect of atmospheric turbulence on the propagation characteristics of the SCMV beam. The intensity distribution of the beams moving through atmospheric turbulence is plotted in Figure 4. In comparison with Figure 3, it conveyed to us a clear message that the SCMV beams showed a clear difference in the propagation characteristics in atmospheric turbulence compared to those exhibited in free space. We can see from the intensity distribution diagram that the near-field region was not much affected by turbulence, just as in the free space case. When the transmission distance increased, the center was no longer a completely dark void. Instead, a ring-shaped cavity and a spectral density peak in the middle indicated that the beam's intensity was concentrated at its center and that the beam profile gradually expanded, wherein as the distance rose, the brightness around the ring-shaped cavity became progressively darker.

Figure 5 shows the transverse distribution of the SCMV beam spectral density for $q = 2$ in atmospheric disturbance with different values of parameters $\alpha$ and $\widetilde{C}_n^2$ at the plane at distance $z = 100$ m. We can observe that when the turbulence parameter values changed, so did the distribution of the spectral density. Comparing Figure 5a–d, Figure 5b–e, and Figure 5c–f, we found that the spectral intensity converged significantly toward the center when $\widetilde{C}_n^2$ was larger. In particular, when $\alpha = 3.10$, the atmospheric turbulence had the greatest effect on the spectral density distribution, and the beam intensity was almost entirely concentrated in the center.

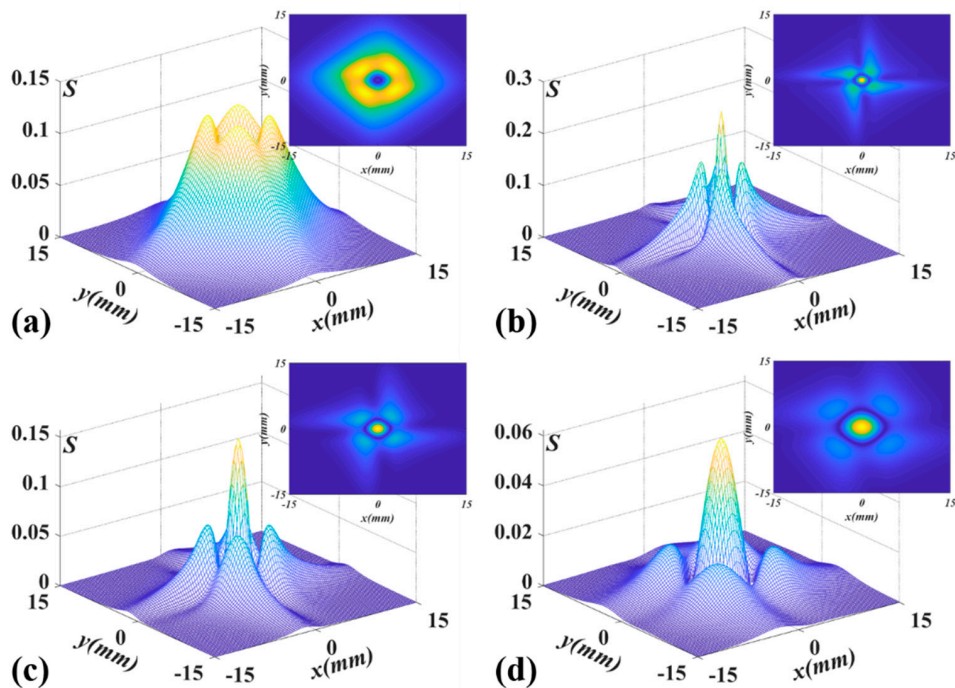

**Figure 4.** The distribution of intensity as the beam travelled through atmospheric turbulence, with $\alpha = 3.667$ and $\widetilde{C}_n^2 = 10^{-12} m^{3-\alpha}$. (**a**) $z = 50$ m; (**b**) $z = 100$ m; (**c**) $z = 150$ m; (**d**) $z = 250$ m.

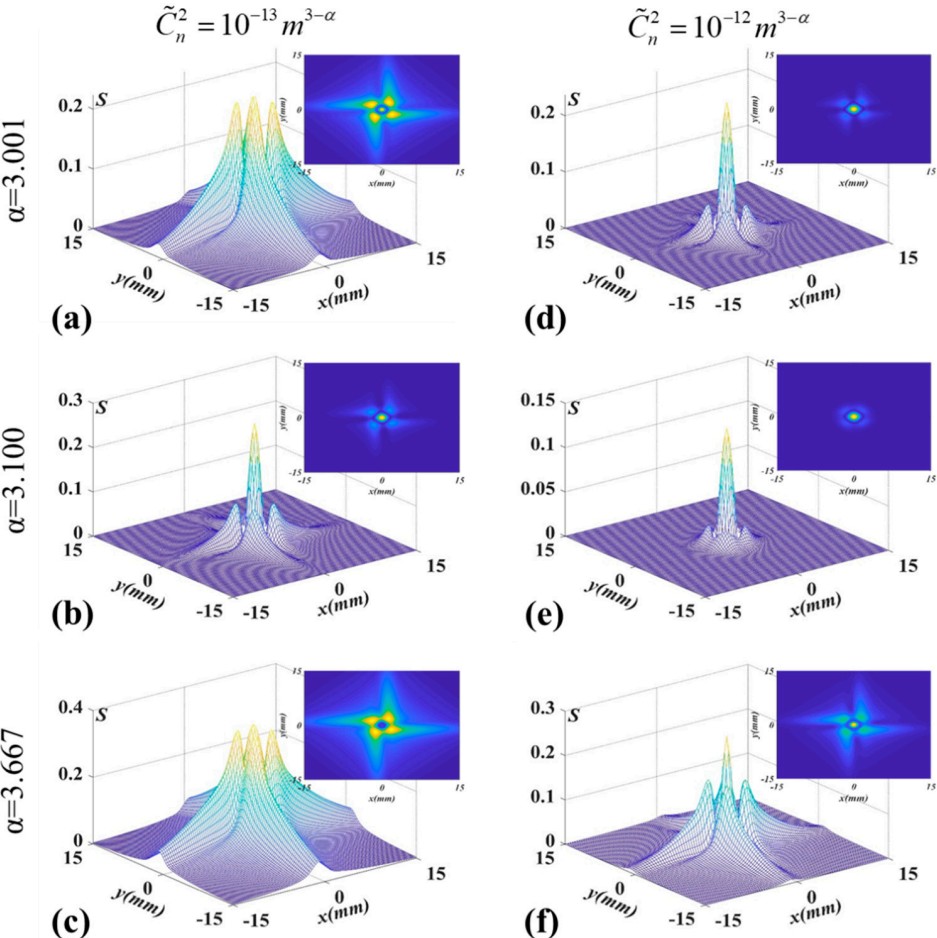

**Figure 5.** The effect of different parameter values of $\alpha$ and $\widetilde{C}_n^2$ on transverse spectral at a distance $z = 100$ m in atmospheric disturbance.

## 5. Conclusions

In this study, we structure a kind of novel vortex light source model in which the degree of coherence is the product of two sinc functions by depending on the $q$th variance of the position coordinates of two points in the source plane. The distribution of light intensity was visualized by numerical simulation, and the propagation properties of light beam in turbulent space were analyzed. The result shows that the radiation field of the light source propagated in free space and exhibited a distinctive self-focusing effect when $q$ was even. We also found that when the power value $q = 2$, the lateral profile of the beam's spectral density was a hollow dark pattern surrounded by four light points in free space. With the distance z from the source plane gradually increasing, the intensity of the beam converged to the center, and this self-focusing effect was most obvious at $z = 100$ m. Then, the beam gradually expanded. When the beams were transmitted in a non-Kolmogorov turbulent medium, it still had a hollow distribution of intensity and was able to remain stable in the region close to the source plane. With increasing distance, however, this dark hollow pattern was disrupted by atmospheric turbulence. A peak of light intensity appeared in the center of the dark hollow where the original light intensity was zero. The effect of turbulence on the light field was analyzed by setting different values of refractive index structural parameters $\widetilde{C}_n^2$ and power spectrum parameters $\alpha$ for atmospheric turbulence. It was found that the effect of turbulence on the light field was more pronounced for larger values of $\widetilde{C}_n^2$. When $\alpha$ was 3.10, the dark hollow structure was most severely disrupted, and the light intensity was almost entirely concentrated in the center of the beam. Our results have potential applications and theoretical guidance for particle manipulation and materials processing with light beams.

**Author Contributions:** Conceptualization, Z.M.; methodology, Z.M. and G.Z.; investigation, J.W.; writing—original draft preparation, X.S. and Y.M.; writing—review and editing, Y.M. and Z.M.; funding acquisition, G.Z.; validation, Z.M. All authors have read and agreed to the published version of the manuscript.

**Funding:** This research was funded by National Natural Science Foundation of China (11974107) and Zhejiang Provincial Natural Science Foundation of China (LY23A040006).

**Institutional Review Board Statement:** Not applicable.

**Informed Consent Statement:** Not applicable.

**Data Availability Statement:** Data underlying the results presented in this paper are not publicly available at this time but may be obtained from the authors upon reasonable request.

**Conflicts of Interest:** The authors declare no conflict of interest.

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
