# Peer review of "High-Order Sinc-Correlated Model Vortex Beams"

_photonics, doi:10.3390/photonics10050550_

Round 1
Reviewer 1 Report
Dear Editor,
For the manuscript numbered photonics-2323969 entitled “High-order Sinc-correlated model vortex beams” by author Zhangrong Mei.
The above quote manuscript has reported a new partially coherent vortex source model, in which the spatial correlation function is a sinc function on the difference from the n-th power of the coordinates of two points of the source field, it is named as the High-order Sinc-correlated model vortex (SCMV) beam. The propagation formulas of the cross-spectral density (CSD) function for SCMV beams in atmospheric disturbances are derived, and the propagation behaviors of the spectral density of the SCMV beams in free space and atmosphere turbulence is investigated comparatively. The results show that the spectral densities of such beams exhibit interesting novel features, which are significantly different from those of the trivial vortex beams. The authors discuss a computational model describing a self-focusing action, which is similar to the results exhibited by the self-focusing vortex beam, and the atmospheric turbulence has the greatest effect on the spectral density distribution, and the beam intensity is almost entirely concentrated in the center when the generalized exponential parameter is 3.100.
The manuscript provides a good overview of the subject before diving in, is moderate organized and logically consistent. I think this article can be published after major revisions. Before publication, it is recommended to address the article in terms of content as below:
(1), The manuscript should explain the importance of a Sinc-correlated model vortex beam propagating through atmospheric turbulence compared with other classical beams. For example, can the transmission of such beams in turbulent atmosphere improve measurement accuracy, imaging resolution than the classical beams?
(2), Please check the correctness of the following description in detail: “The CSD function expression can be written as follows by combining Eq. (5) and Eq. (6) into Eq. (4)”.
(3), In Abstract, author states that “We derive the propagating formula of the cross-spectral density (CSD) function for SCMV beams in atmospheric disturbances.”. However, the Eq.(11) is not be solved. So, can it be done when n=1?
(4), I want to know that can Eq. (6) be given directly without Eqs.(1)-(5)?
(5), Please check the formula number of Eqs.(1)-(5) in detail.
Reviewer 2 Report
In this manuscript, the properties of high-order Sinc-correlated model vortex beams in atmospheric disturbances are discussed. It is found the radiation field of the light source propagates in free space and exhibits a distinctive self-focusing effect when n is even. Especially, the effect of turbulence on the light field was analyzed for the case n=2. In my opinion, the paper is clearly written and the result is simple to find. But some issues need the authors to add and address, which may helpful for promoting this work to broadly audience.
1) Why does this self-focusing behavior occur when n is even? Can you explain it in a physical or mathematical perspective?
2) The effect of turbulence on the light field is only listed for the case of n=2 in Figs 2-4. Why do you choose n=2? Does this turbulence have the same impact when n is even? What about the case n is odd?
3) A higher order vortex beam generally appears to be unstable when it propagates in reality. Why do you choose the higher order sinc-correlated model vortex beams in this manuscript? What is the advantage and the application of the higher order sinc-correlated model vortex beams? You can add the content in the introduction.
Reviewer 3 Report
Review comments (Manuscript ID: photonics-2323969):
In this paper, the authors propose a new partially coherent vortex source model and present the analytical propagation expression of cross-spectral density (CSD) function of High-order Sinc-correlated model vortex (SCMV) beam in turbulent atmosphere. Meanwhile, the propagating behavior of the CSD of SCMV beam in free space and atmospheric turbulence is investigated. This work is original and undoubtedly contains new information and the results are interesting, but I have some questions and comments:
1. All the equation serial numbers quoted in the manuscript are wrong, such as “The CSD function expression can be written as follows by combining Eq. (5) and Eq. (6) into Eq. (4)” in lines of 83 and 84, “The CSD represented by Eq. (7) fulfills the nonnegative definiteness condition” in line of 87, “When Eq. (7) is substituted into Eq. (8),” in line of 106, et al.
2. In the manuscript, the Eq. (14), which is used to calculate the spectral density of SCMV beam propagating through free space and the non-Kolmogorov turbulence is not concretely presented, though it is the most important in this paper.
3. The title of this manuscript is “High-order Sinc-correlated model vortex beams”, but the authors just discuss the propagating characteristics of the SCMV with topological charge l=1. Therefore, the authors should supplement the results of high-order SCMV with different topological charge in the free space and the turbulent atmosphere. In addition, the propagation properties of SCMV with different source parameters values and different propagating conditions should be provided.
4. The authors only describe the simulation results in the manuscript. It would be of interest for readers if the physical explanations or theoretical analysis for the modeling results are added.
5. The authors do not discuss much about the applications and impact of the results. The possible impact mentioned there is rather weak.
6. The languages in the manuscript need to be polished seriously.
Reviewer 4 Report
The work entitled "High-order Sinc-correlated model vortex beams" is an interesting work that proposes a partially coherent vortex source model. However, there are several issues that need to be addressed before it gets accepted for publication.
1) The manuscript is very difficult to read, it requires a thorough grammar review. Example, see page 2, line 79.
2) Section two needs more work to better describe the model. The mathematics' nomenclature needs to be expressed in a clear and transparent way. For instance, what is the relationship of the superindex n in the delta functions of equation 4, and the subindex n in equation 8. The authors should breakdown this section into two at least, and have the turbulance part apart from the vortex model; also a better explanation of the turbulance approximation is necessary, why turbulance is important?. Also eachj index should be better described, for instance the meaning of l (topological charge) does not come out till the results section.
3)The objective of this research is not clear, even though authors declare that these beams inevitably interact with an external medium because the quality of the light beams will be decreased, What features of these novel beams are important and why? Authors should discuss more such importance.
Round 2
Reviewer 1 Report
The revisions are correct. It is recommended that this article can be published in photonics.
Author Response
We thank the reviewers once again for your helpful remarks in improving the quality of the manuscript.
Reviewer 2 Report
The content of the manuscript has improved a lot. Please pay attention the spell of the word. For example, the word "rusult" in the line 147 needs to be corrected. The figures in the line 180, 184, 186 seem redundant.
Author Response
We have carefully checked the whole manuscript and corrected grammar and spelling errors. We thank the reviewers once again for your helpful remarks in improving the quality of the manuscript.
Reviewer 3 Report
There are some formatting errors in the arrangement of the text and some redundant figures on 7 and 8 pages. I am recommending publication, but I hope that the English of this manuscript needs to be further polished.
Author Response
We have carefully checked and modified the format of the text and tried our best to improve the language expression. We thank the reviewers once again for your helpful remarks in improving the quality of the manuscript.
Reviewer 4 Report
After a careful second review, the authors have complied with all my questions and concerns. However, there are still some typos in the manuscript. For instance, see page 5, line 147, please review again!
Author Response
We have carefully checked the expression of words and expressions in this article again to ensure that there are no mistakes. We thank the reviewers once again for your helpful remarks in improving the quality of the manuscript.